# Hydrothermal Synthesis of Zinc Oxide Nanoparticles Using Different Chemical Reaction Stimulation Methods and Their Influence on Process Kinetics

**DOI:** 10.3390/ma15217661

**Published:** 2022-10-31

**Authors:** Tomasz Strachowski, Magdalena Baran, Marcin Małek, Robert Kosturek, Ewa Grzanka, Jan Mizeracki, Agata Romanowska, Stefan Marynowicz

**Affiliations:** 1Lukasiewicz Research Network–Institute of Microelectronics and Photonics IMIF, Research Group of Graphene and Composites, al. Lotników 32/46, 02-668 Warsaw, Poland; 2Faculty of Civil and Engineering and Geology, Research Laboratory of WIG, Military University of Technology, ul. Gen. Sylwestra Kaliskiego 2, 00-908 Warsaw, Poland; 3Faculty of Mechanical Engineering, Institute of Robots & Machine Design, Military University of Technology, ul. Gen. Sylwestra Kaliskiego 2, 00-908 Warsaw, Poland; 4Institute of High Pressure Physics PAS, ul. Sokołowska 29/37, 01-141 Warsaw, Poland

**Keywords:** zinc oxide nanoparticles, chemical reactors, hydrothermal synthesis

## Abstract

The aim of this work was to study the effect of the applied chemical reaction stimulation method on the morphology and structural properties of zinc oxide nanoparticles (ZnONPs). Various methods of chemical reaction induction were applied, including microwave, high potential, conventional resistance heater and autoclave-based methods. A novel, high potential-based ZnONPs synthesis method is herein proposed. Structural properties–phase purity, grain size–were examined with XRD methods, the specific surface area was determined using BET techniques and the morphology was examined using SEM. Based on the results, the microwave and autoclave syntheses allowed us to obtain the desired phase within a short period of time. The impulse-induced method is a promising alternative since it offers a non-equilibrium course of the synthesis process in an highly energy-efficient manner.

## 1. Introduction

Zinc oxide nanoparticles are known for their broad range of applications. They are widely used in the pharmaceutical industry due to their antifungal properties. They are also used as an ingredient in toothpastes and as a material for dentistry. In the paint industry, ZnONPs are used in the production of white pigment, so-called zinc white. The ceramics industry uses zinc oxide as an ingredient in ceramic masses, glazes and fluxes. Zinc oxide nanoparticles are used as a catalyst in organic synthesis. In the electronics industry, ZnONPs are used to produce semiconductors [1,2,3,4,5], varistors [6,7,8,9,10,11,12], sensors [13,14,15,16,17,18] and transistors [19,20,21,22,23,24]. Due to the wide energy gap, 3.37 eV zinc oxide is used as a luminescent material in optoelectronics [25,26,27,28,29,30]. Application of superfine (nanometric) ZnONPs and thin films in the production of coatings for UV protective glasses and polymer coatings have been in use for a long time [31,32,33].

ZnONPs can be obtained by several methods. These include: the oxidation of metallic zinc [34,35,36], sol-gel [37,38,39,40], co-precipitation [41,42,43,44], gas condensation [45,46,47,48], mechano-chemical method [49,50,51,52], solvothermal method [53,54,55,56] and hydrothermal method [57,58,59,60,61,62,63,64]. Among the known methods for producing nanomaterials, preparation of zinc oxide from selected zinc salts and mineralizer may be an example of a simple synthesis resulting in a ceramic raw material. The resulting zinc hydroxide can be transformed by calcination or heated in water at a pressurized vessel (hydrothermal synthesis). Hydrothermal synthesis is carried out in a closed vessel in order to avoid contamination of the surroundings with nanopowders, and is carried out at lower temperatures than calcination, which prevents sintering of the nanopowders. From the literature and our own research, the correct structure of zinc oxide in the hydrothermal reaction is formed at the temperature not exceeding 100 °C at a pressure of ca. 0.15 MPa [65].

Several works have demonstrated [66,67,68,69,70] the benefits of using microwave reactors in hydrothermal syntheses. Above all, the use of microwaves provides uniform heating of the solution. Moreover, the high reaction speeds of inorganic syntheses reactions in a microwave field are often discussed in terms of the so-called specific microwave effect. Many chemical syntheses under the influence of microwaves proceed more efficiently and faster as compared to traditional methods [66,67,68,69,70]. Similarly, fast and in-volume heating of reaction substrates can be ensured with the application of current flow [57,66]. Our previous study revealed that ZnONPs can also be obtained using high voltage pulses [71]. Thus, in terms of scalability of the process, the choice of a proper synthesis method has to be considered. A systematic comparative study of hydrothermal ZnONPs synthesis methods is hereby conducted [72,73,74].

## 2. Materials and Methods

### 2.1. Materials

The following substances were used for hydrothermal synthesis:Zinc chloride–ZnCl_2_ (99.9%, anhydrous)–Sigma-AldrichPotassium hydroxide–KOH (pure)–Chempur, Poland

### 2.2. Methods

#### 2.2.1. Microwave Reactor

The obtained zinc hydroxide suspension was poured into reaction vessels. ZnONPs were synthesized via hydrothermal synthesis in a microwave field using a microwave reactor (Figure 1) (ERTEC, Poland). This reactor consisted of a pressure chamber, a PTFE reaction vessel with a sealed lid and a pressure diaphragm as a burstable pressure fuse. The chamber was made of acid-resistant steel and cooled with water. The occlusion was of a bayonet type, and pressure measurement was controlled by measuring the deflection of disc springs. Electromagnetic field in the pressure chamber was generated using waveguides. The experimental setup was controlled using a computer and dedicated software—it allowed for the adjusting of power thresholds and pressure limits, along with temperature and pressure control throughout the process [66,67,68,69,70,75].

The experiment in the microwave reactor proceeded as follows: The prepared solution was poured into a PTFE reaction vessel with a capacity of 110 mL. Several process parameters required optimization-process duration (heating and cooling time), maximum temperature and the pressure, at which the process needed to proceed. An important parameter was also the microwave power. Overall hydrothermal synthesis using the microwave reactor process did not exceed 30 min. After completion of the processes, the product was pressure filtered and washed repeatedly with distilled water. The purified material was then vacuum dried (70 °C/24 h).

#### 2.2.2. Autoclave

The autoclave (Figure 2) was heated by electric heating spiral heating coils placed in the walls of the vessel. The capacity of the reaction vessel was 2 L. Inside of the autoclave there was a PTFE insert, in which the reaction substrates were placed. With this method, it took only 15 min to obtain the required reaction temperature. The total reaction time reached 3 h due to the thermal inertia of the entire system. During these experiments, the autoclave with a PTFE reaction vessel (CORTEST, Willoughby, OH, USA) was used.

#### 2.2.3. Impulse Reactor

Pulsed reactors for chemical syntheses are little known in industrial practice and literature. The use of electrodes immersed in solution is mostly limited to galvanic processes in coating fabrication. Meanwhile, independently from classical electrochemistry, applications of pulsed fields are being developed, generating high electric field strengths in liquids. One of the applications of pulsed reactors is the sterilization beverages by the PEF (pulsed electric field) method. The use of electric pulses to conduct chemical reactions has been patented [71]. In our experiments, short electric pulses of between 50 ns, 20 ms of 0–30 kV range and energies up to 120 J were induced at a frequency of 1 Hz between electrodes immersed in the reaction substrates (Figure 3). The use of this type of excitation in a chemical reactor enabled us to obtain the effect of volumetric excitation, similar to the microwave reactor method.

#### 2.2.4. ERTEC Reactor with Meander Heater and Volume Type Joule

In order to compare the different ways of providing energy required to perform a chemical reaction, the microwave reactor was modified. The reactor was suitably adapted for heating using the Joule phenomenon or conventional heating by means of an electric immersion heater in the shape of a meander. The vessels and the shape of the reactor remained unchanged.

The heaters of both types were made in such a way as to provide similar power values as in the case of microwave heating. To stabilize the power level, electronic regulators used in the case of microwave power corresponded to a constant average value of the magnetron current. A power level of 300W was maintained in all experiments with an accuracy better than 5%.

Heater surfaces, in order to increase their resistance to corrosion, were covered with a golden layer of 30 μm thickness. Moreover, the reaction vessel was equipped with a special PTFE lid, through which a system of two stainless steel coaxial tubes was passed to provide the electrical connection. The middle pipe, in addition to being an electrical conductor, acted as a screen for the fiber optic temperature sensor (Nortech, Virginia, MN, USA). Pressure measurements were carried out both by measuring the deflection disk springs according to DIN 2993 installed in the original reactor, as well as by measuring the deflection of the plate springs using a classical membrane pressure transducer (ZAP S.A PL, PN-5/M/42057, Ostrów Wielkopolski, Poland) connected to the lid of the reaction vessel by means of a capillary tube. Figure 4 shows a schematic of the heaters used in the zinc oxide nanoparticles synthesis.

#### 2.2.5. XRD Analysis

A Siemens D-5000 X-ray diffractometer (Siemens-Bruker Corporation, Karlsruhe, Germany) was used to determine the phase composition, crystallographic lattice and to identify the coordinates of atoms in the elementary cells (by the Rietveld method) [76,77,78].

#### 2.2.6. Density and Specific Surface Area (BET)

Specific surface of the obtained materials was determined with BET technique (Gemini 2360, Micromeritics, Norcross, GA, USA). Powder density was calculated/determined with the pycnometric method and was an important indicator of obtained material purity. Specific surface area and density measurements allowed for determination of the powder grain size, following the formula:(1)$$d=\frac{6}{\rho \cdot S}*1000$$
where *d*—grain size [nm], *ρ*—powder density [g/cm^3^], *S*—powder specific surface area [m^2^/g], the number 6 is taken for nanoparticles with a spherical shape.

#### 2.2.7. SEM Analysis

Microstructure studies of nanoparticles were performer using scanning microscopy with a Zeiss LEO1530 (Siemens-Bruker Corporation, Germany) equipped with a Zeiss Gemini column.

## 3. Experiments and Results

All solutions were prepared with the same route, only the method of chemical reaction stimulation was modified. Aqueous solutions of zinc chloride and potassium hydroxide were mixed, leading to an ion-exchange reaction:ZnCl_2_ + 2KOH + H_2_O → Zn(OH)_2_ + 2KCl + H_2_O(2)
Zn(OH)_2_ + KCl + H_2_O → ZnO + KCl + 2H_2_O(3)

After each synthesis, the product was filtered and rinsed several times with distilled water to remove excessive ions, following with isopropyl alcohol rinsing to reduce the agglomeration. Afterwards, the product was vacuum dried at 100 °C for 24 h. The dried product was subsequently characterized with XRD analysis, BET method, density measurements and SEM.

Table 1 presents the results of measurements of the specific surface of powders obtained in the studied reactors. The grain size measured by BET method and on the basis of XRD is also presented.

Figure 5 shows the results of the XRD analysis of powders obtained under different conditions. Additional phases, such as hydrozincite (Zn_5_(CO_3_)_2_(OH)_6_) and simonkolleite (Zn_5_(OH)_8_Cl_2_·H_2_O) can be noticed. A single-phase product (ZnO) was obtained using a microwave reactor and an autoclave stimulation method. The blue circle indicates additional phases.

From the SEM images (Figure 6), it can be seen that depending on the applied chemical reaction technique applied the obtained zinc oxide nanoparticles is characterized by varied morphology. We can distinguish several morphological forms formed as a result of agglomeration. The product obtained in the microwave reactor revealed a cubic structure, while in the autoclave pictured lamellar structure. Using a meander-type heater the product obtained had a needle and plate structure. The product of Joule heating was very similar–plates and spheres. The product from the pulse reactor—flowers—also had an interesting morphology. It should be noted that the size of these structures were much larger than the size of crystals determined by XRD methods. It follows that these are agglomerates built from many grains.

## 4. Discussion

As can be seen from Table 1, all the synthesis methods studied lead to the formation of powders with medium grains in the range of 30–50 nm. All of the synthesis reactions were conducted at temperatures below 100 °C with the exception of synthesis in an autoclave, where the temperature reached 200 °C. In addition, when heated using the volume effect, Joule effect and electric heating, two- or three-phase products with simonkolleite or hydrozincite were formed. When heating by means of heaters the product was strongly contaminated due to the corrosion of electrodes in an alkaline environment. Only microwave heating and synthesis in an autoclave resulted in a pure phase. In the case of the pulse reactor, traces of an unidentified phase were found. The purity of the product obtained in the case of microwave heating and in the autoclave clearly results from the electrodeless way of heating. The microwave energy penetrates through a clean Teflon vessel directly into the reaction substrates, heating them quickly and evenly. In an autoclave with a Teflon insert, where the energy is supplied from outside the reactor, the heating process is much slower but also occurs without generating contamination. In the case of stimulation reaction with high voltage electrical impulses, a pure product is probably produced because the average reaction temperature is very low, the lowest of all the measured during the tests, so the electrode material was not easily dissolved. In addition, the erosion of the electrodes in strong electrolytes under pulsed current is always less than with direct current.

Some of the key parameters to the growth of crystallites are the process temperature, pressure, chemical reagents and process conditions, such as the use of different techniques for stimulating chemical reactions. Even a small change in technique causes huge changes both in the growth process itself (e.g., decomposition of the precursor) and in the parameters of the resulting material. Such material can take on various morphologies, such as flakes, columns or beads, which can form agglomerates (Figure 7). For zinc oxide particles, the correct morphology is a hexagonal structure.

Another important parameter that affects the morphology of zinc oxide is the environment in which the synthesis is carried out, namely the pH (Figure 8).

Scanning microscopy images revealed that the reaction product is agglomerated in the form of needles, petals or flowers. These structures are made up of tiny crystallites of ZnO, which are attracted to each other in the suspension or during rinsing. Morphology formation by different synthesis methods requires further investigation. The difference in the morphology of the resulting products may be due to the heating method. In the microwave reactor and in the autoclave, heating was supplied from outside. In both cases, the heating medium was not in direct contact with the solution. As for the synthesis in the pulse reactor and during meander and Joule type heating, the heating element was immersed in the solution. The direct contact of the heating element with the solution can affect the morphology and purity of the obtained product. From the analyses, it can be seen that traces of simonkolleite and hydrozincite were obtained in addition to phase-pure zinc oxide.

The morphology of zinc oxide has a significant impact on its physicochemical properties, such as photocatalytic, optical, electrical and antibacterial. As a result, it can be used in many industrial fields.

The shortest reaction times and the lowest process temperatures were observed when stimulated with high voltage pulses. Investigating the nature of this surprising discovery encourages the continuation of experimental work in the field of impulse synthesis of zinc oxide nanoparticles.

A novel method for the synthesis of oxide nanoparticles in a pulsed reactor has important advantages:The product has high purity and density close to the theoretical value-only traces of an unidentified phase were found, which should be minimized by the choice of optimal electrode material;The relatively high specific surface area of the powder allows obtaining zinc oxide nanoparticles at an unprecedentedly low temperature of approx. 25 °C.

Joule heating, on the other hand, is attractive because of its simplicity–energy is saved for heating the vessel and the process is interrupted when current flow is interrupted. Patent claims have been filed for both synthesis methods [71]. Contamination of the nanoparticles has been observed as a result of corrosion of the electrodes, and deposition of the nanoparticles on the electrodes, which sometimes altered the current and voltage in the reaction substrates during synthesis. Corrosion and unfavorably large temperature gradients occurred during heating with a conventional resistance heater with a meander shape.

As can be seen from the above, the most promising from the scientific and practical point of view is high-voltage pulse synthesis, because on the one hand, the nature of this phenomenon requires investigation, while on the other hand, the results of laboratory syntheses indicate the possibility of direct implementation of this technique to industrial practice. However, any laboratory process must undergo pilot-scale verification before it reaches industry. From the point of view of production of nanoparticles of the highest quality and purity, the most promising is the hydrothermal method with microwaves as the energy source.

## 5. Conclusions

Zinc oxide nanoparticles obtained by hydrothermal synthesis using several reactors, in which the energy for the reaction was supplied by microwaves, electric current, volumetric Joule heating and via high voltage pulses and heating the entire autoclave. In all of the conducted syntheses with the exception of the classical autoclave, the temperature process did not exceed 100 °C. In the autoclave the temperature reached 200 °C. Due to the high thermal inertia of the autoclave its application to fast chemical syntheses is impossible.

To our knowledge, we are the first team to obtain zinc oxide nanoparticles as a result of syntheses induced by high voltage pulses. As a result of the application of the pulses, the average reaction temperature in did not exceed 50 °C in any of our experiments. Syntheses driven by the Joule heating method encountered the obstacle of corrosion of the electrodes and deposition of the reaction product on them.

It was found that nanoparticles with phase composition nanopowders with phase compositions closest to pure ZnO were obtained during microwave syntheses and in a classical autoclave. The powders produced in other reactors were characterized.

In addition to the zinc oxide content, powders produced in other reactors were characterized by the presence phases—simonkolleite and hydrozincite. Application of the microwave reactor allowed us to shorten the reaction time by several times and, at the same time, to obtain the purest product. It is this type of reactor that we will develop for the implementation experimental production line of zinc oxide nanoparticles.

## Figures and Tables

**Figure 1 materials-15-07661-f001:**
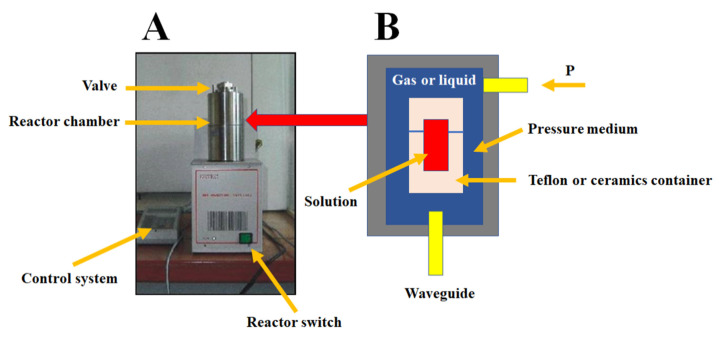
Microwave reactor: (**A**) overall view, (**B**) schematic.

**Figure 2 materials-15-07661-f002:**
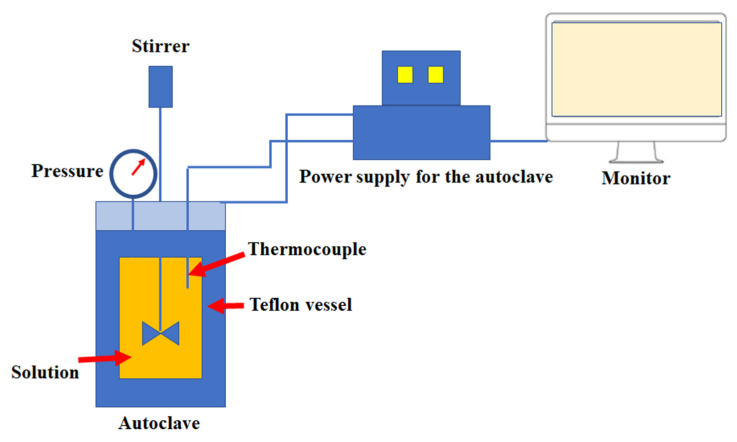
Autoclave system diagram.

**Figure 3 materials-15-07661-f003:**
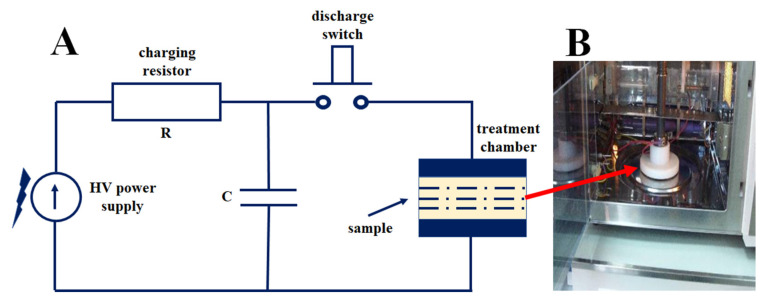
Impulse reactor: (**A**) schematic; (**B**) inside the reactor.

**Figure 4 materials-15-07661-f004:**
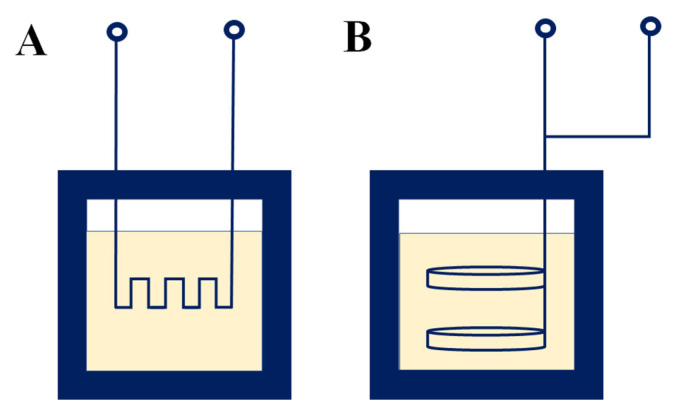
Schematic of the heaters: (**A**) meander heater; (**B**) type Joule heater.

**Figure 5 materials-15-07661-f005:**
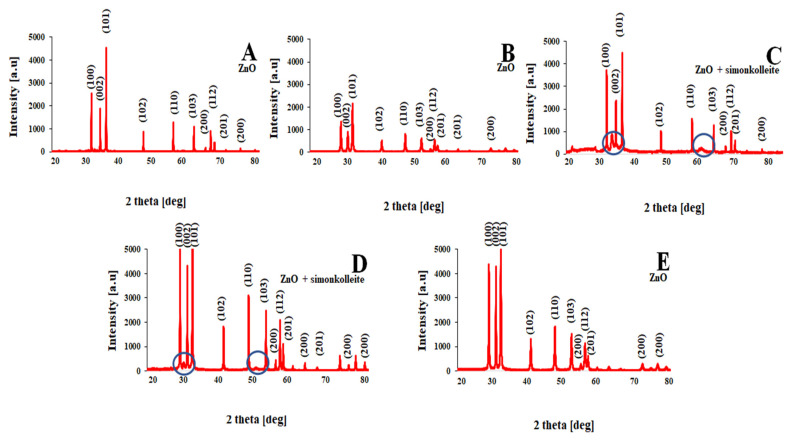
XRD analysis: (**A**) microwave reactor; (**B**) autoclave; (**C**) meander heater; (**D**) Joule type heater; (**E**) impulse reactor.

**Figure 6 materials-15-07661-f006:**
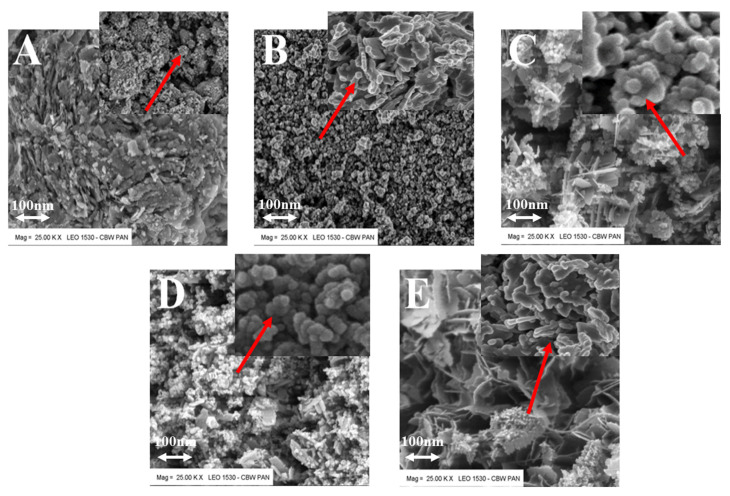
SEM pictures: (**A**) microwave reactor; (**B**) autoclave; (**C**) meander-type heater; (**D**) Joule type heater; (**E**) impulse reactor.

**Figure 7 materials-15-07661-f007:**
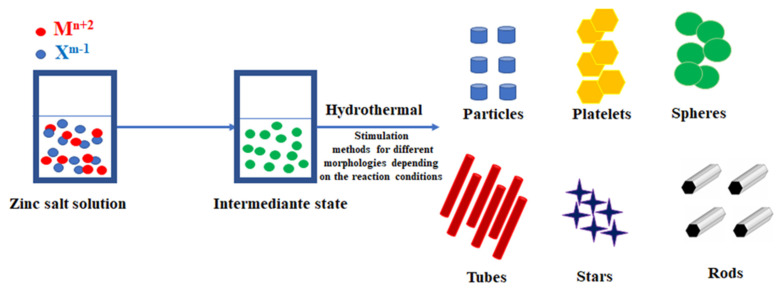
Schematic illustration for the morphology engineering of zinc oxide nanoparticles.

**Figure 8 materials-15-07661-f008:**
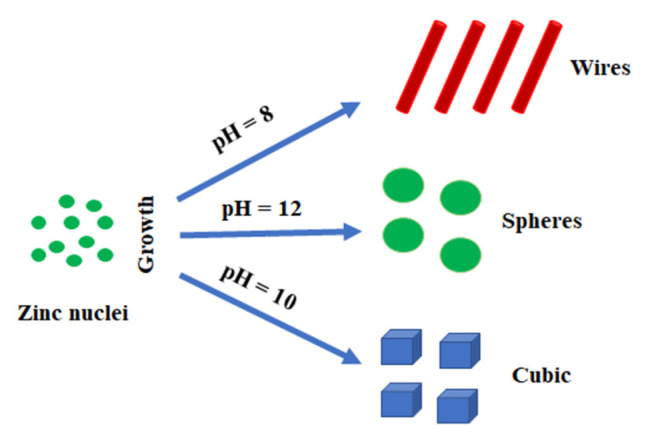
pH influence for morphology of zinc oxide nanoparticles.

**Table 1 materials-15-07661-t001:** Results of analyses the zinc oxide nanoparticles.

Stimulaton Methods	Time [min]	Pressure [MPa]	Temperature T_s_/T_e_ [°C]	Shape	Color	BET [nm]	Average Crystallite Size [nm]	Density [g/cm^3^]	Phases
Microwave reactor	3	1	25/80	Cubic	White	37	25	5.60 ± 0.02	ZnO
Electric autoclave	15	20	25/200	Plates	White	39	24	5.60 ± 0.03	ZnO
Meander-type heater	3	1	25/78	Needles/plates	Gray	32	20	5.52 ± 0.02	ZnO + simonkolleite
Joule-type heater	4	2	26/88	Plates/spheres	Beige	40	32	5.50 ± 0.04	ZnO + simonkolleite
Impulse reactor	0.3	1	25/50	Flowers	White	46	40	5.60 ± 0.05	ZnO + unknown phase

T_s_ starting temperature; T_e_ ending temperature.

## Data Availability

Not applicable.

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
