# Peer review of "Hydrothermal Synthesis of Zinc Oxide Nanoparticles Using Different Chemical Reaction Stimulation Methods and Their Influence on Process Kinetics"

_materials, 2022, doi:10.3390/ma15217661_

Round 1

Reviewer 1 Report

The aim of this work was to study the effect of the applied chemical reaction stimulation method on the morphology, phase composition and structure of ZnONPs. All methods were done in aqueous media, starting from ZnCl2 and KOH as mineralizer. Some interesting results have been obtained by comparing specific area, phases detected, mean crystallite sizes.

Please consider the following comments:

- The title "Hydrothermal synthesis of zinc oxide nanoparticles using different chemical reaction stimulation methods" is ambiguous. In fact, the reaction mechanism is the same and the effect is to enhance the kinetics of the process.

- The phrase: From the literature and our own research the correct structure of zinc oxide in the hydrothermal reaction is formed at the temperature not exceeding 100°C (ca. 120°C) at a pressure of ca. 0.15 MPa. Since 120 deg.C> 100 deg.C, the statement is not correct. Please re-formulate.

- Table 1 presents the results of measurements of the specific surface of powders. Actually, the table presents also other results. Please reformulate the title of the table accordingly.

- In the same table 1 as parameters are mentioned Tp and Tk. Please explain their meaning.  Also, the column XRD [nm] should be rename as mean crystallite size.

- In figure 5 please consider adding [hkl] indexing of peaks

- The statement "documented quality" needs more experimental data, for example mot only a qualitative, but a quantitative phase composition.

- The scalability of a process may be assessed on many other aspects, including reproducibility, investments needed, economic and environmental aspects. Therefore, the statement "the results of laboratory syntheses indicate the possibility of direct implementation of this technique to industrial practice" must be reconsidered, any laboratory process must undergo at least a pilot scale verification. 

Author Response

REVIEWER 1 (Round 1)

Comments and Suggestions for Authors

The aim of this work was to study the effect of the applied chemical reaction stimulation method on the morphology, phase composition and structure of ZnONPs. All methods were done in aqueous media, starting from ZnCl2 and KOH as mineralizer. Some interesting results have been obtained by comparing specific area, phases detected, mean crystallite sizes.

Please consider the following comments:

- The title "Hydrothermal synthesis of zinc oxide nanoparticles using different chemical reaction stimulation methods" is ambiguous. In fact, the reaction mechanism is the same and the effect is to enhance the kinetics of the process.

Title have changing „Hydrothermal synthesis of zinc oxide nanoparticles using different chemical reaction stimulation methods and their influence on process kinetics

- The phrase: From the literature and our own research the correct structure of zinc oxide in the hydrothermal reaction is formed at the temperature not exceeding 100°C (ca. 120°C) at a pressure of ca. 0.15 MPa. Since 120 deg.C> 100 deg.C, the statement is not correct. Please re-formulate.

This sentence has been corrected in the text. – LINES 54-56

- Table 1 presents the results of measurements of the specific surface of powders. Actually, the table presents also other results. Please reformulate the title of the table accordingly.

The title of the table is entered correctly. Corrections have been added to the Table 1.

- In the same table 1 as parameters are mentioned Tp and Tk. Please explain their meaning.  Also, the column XRD [nm] should be rename as mean crystallite size.

There has been a mistake. Polish markings were entered. They have already been corrected in the Table 1.

- In figure 5 please consider adding [hkl] indexing of peaks

Peaks [hkl] have been added on the Figure 5

- The statement "documented quality" needs more experimental data, for example mot only a qualitative, but a quantitative phase composition.

Information included in the text.

- The scalability of a process may be assessed on many other aspects, including reproducibility, investments needed, economic and environmental aspects. Therefore, the statement "the results of laboratory syntheses indicate the possibility of direct implementation of this technique to industrial practice" must be reconsidered, any laboratory process must undergo at least a pilot scale verification. 

Information included in the text. LINES 263-264

Reviewer 2 Report

The manuscript is focused on the synthesis and characterization of ZnO powders by various methods of chemical reaction induction. Even though the work is interesting, the observed phenomena are presented and discussed with unclear explanations. More data is needed for increasing the authenticity and clarity of the results. After that, it is publishable on the journal of materials after mayor revisions.

1. The authors need to check if your manuscript satisfies the Journal's format. In this context, the English should be improved and some punctuation or grammatical mistakes should be revised.

2. The authors should choose better XRD patterns (the must have the same intensity to be able to compare them) and these should be indexed with (h k l) values.

3. The growth mechanism for formation of ZnO particles should be clearly discussed based on chemistry concepts. The authors should consider that agglomerates can be composed of mechanically bonded particles and mostly can be broken down.

4. In order to corroborate the size of individual particles, whose sizes are in nano range, more characterization data is needed for this work, it includes high mag SEM, AFM and HRTEM. After that, please contrast and compare the key results (high mag SEM, AFM and HRTEM) in detail among the various methods of chemical reaction induction applied in this work. Please, show these nanoparticles with arrows on the new figures.

5. The authors state “…zinc oxide nanoparticles is characterised by varied morphology. We can distinguish several morphological forms formed as a result of agglomeration.”. Taking into account that agglomerates can be composed of mechanically bonded particles and mostly can be broken down, so What is the right morphology of the ZnO particles?

6. Physically-joined particles are agglomerates not micro or nanoparticles and by, for exp., simple ultrasonication in an appropriate solution, the agglomerates can be disaggregated. If not, then one cannot speak of nanoparticles but rather microparticles with nanograins. How do the authors relate agglomerates with the occurrence of nanoparticles?

7. Authors should include the following references in the introduction part for more readable, relevant to different methods used to synthesize ZnO particles and its property-applications: (a) Karam, S. T., Abdulrahman, A. F. (2022, August). Green Synthesis and Characterization of ZnO Nanoparticles by Using Thyme Plant Leaf Extract. In Photonics (Vol. 9, No. 8, p. 594). MDPI; (b) Al-She'irey, A. Y., Balouch, A., Mawarnis, E. R., Roza, L., Rahman, M. Y. A., Mahar, A. M. (2022). Effect of ZnO seed layer annealing temperature on the growth of ZnO nanorods and its catalytic application. Optical Materials, 131, 112652; (c) Rojas-Chávez, H., Miralrio, A., Hernández-Rodríguez, Y. M., Cruz-Martínez, H., Pérez-Pérez, R., Cigarroa-Mayorga, O. E. (2021). Needle-and cross-linked ZnO microstructures and their photocatalytic activity using experimental and DFT approach. Materials Letters, 291, 129474.

8. What is roughly the ratio between the amount of the nanoparticles and agglomerates? If the amount of nanoparticles are much larger than that of agglomerates, then it is possible that some properties of nanoparticles suppress or overcome that of agglomerates. Consequently, considering the broad range of ZnO applications, the authors should consider evaluate some ZnO property related to its particle size.

Author Response

REVIEWER 2 (Round 1)

The manuscript is focused on the synthesis and characterization of ZnO powders by various methods of chemical reaction induction. Even though the work is interesting, the observed phenomena are presented and discussed with unclear explanations. More data is needed for increasing the authenticity and clarity of the results. After that, it is publishable on the journal of materials after mayor revisions.

  1. 1. The authors need to check if your manuscript satisfies the Journal's format. In this context, the English should be improved and some punctuation or grammatical mistakes should be revised.

The text was checked for language, punctuation and grammar

  1. 2. The authors should choose better XRD patterns (the must have the same intensity to be able to compare them) and these should be indexed with (h k l) values.

Peaks [hkl] have been added on the Figure 5

  1. 3. The growth mechanism for formation of ZnO particles should be clearly discussed based on chemistry concepts. The authors should consider that agglomerates can be composed of mechanically bonded particles and mostly can be broken down.

The information and pictures was included in the text of the manuscript. FIGURE 7 AND FIGURE 8 – LINES 239-250

  1. 4. In order to corroborate the size of individual particles, whose sizes are in nano range, more characterization data is needed for this work, it includes high mag SEM, AFM and HRTEM. After that, please contrast and compare the key results (high mag SEM, AFM and HRTEM) in detail among the various methods of chemical reaction induction applied in this work. Please, show these nanoparticles with arrows on the new figures.

Unfortunately, our team has no access to HRTEM method characterisation. In terms o AFM, it will be impossible for us to realise additional measurements within upcoming weeks.

SEM images include higher magnification sections. The red arrow marks the nanoparticles.

  1. 5. The authors state “…zinc oxide nanoparticles is characterised by varied morphology. We can distinguish several morphological forms formed as a result of agglomeration.”. Taking into account that agglomerates can be composed of mechanically bonded particles and mostly can be broken down, so What is the right morphology of the ZnO particles?

The information was included in the text of the manuscript. LINES 239-245

  1. 6. Physically-joined particles are agglomerates not micro or nanoparticles and by, for exp., simple ultrasonication in an appropriate solution, the agglomerates can be disaggregated. If not, then one cannot speak of nanoparticles but rather microparticles with nanograins. How do the authors relate agglomerates with the occurrence of nanoparticles?

The materials obtained consist of nanoparticles that form agglomerates. LINES 213-214

  1. 7. Authors should include the following references in the introduction part for more readable, relevant to different methods used to synthesize ZnO particles and its property-applications: (a) Karam, S. T., Abdulrahman, A. F. (2022, August). Green Synthesis and Characterization of ZnO Nanoparticles by Using Thyme Plant Leaf Extract. In Photonics (Vol. 9, No. 8, p. 594). MDPI; (b) Al-She'irey, A. Y., Balouch, A., Mawarnis, E. R., Roza, L., Rahman, M. Y. A., Mahar, A. M. (2022). Effect of ZnO seed layer annealing temperature on the growth of ZnO nanorods and its catalytic application. Optical Materials, 131, 112652; (c) Rojas-Chávez, H., Miralrio, A., Hernández-Rodríguez, Y. M., Cruz-Martínez, H., Pérez-Pérez, R., Cigarroa-Mayorga, O. E. (2021). Needle-and cross-linked ZnO microstructures and their photocatalytic activity using experimental and DFT approach. Materials Letters, 291, 129474.

References have been included in the text. LINES 520-527

  1. 8. What is roughly the ratio between the amount of the nanoparticles and agglomerates? If the amount of nanoparticles are much larger than that of agglomerates, then it is possible that some properties of nanoparticles suppress or overcome that of agglomerates. Consequently, considering the broad range of ZnO applications, the authors should consider evaluate some ZnO property related to its particle size.

The information was included in the text of the manuscript.LINES 267-269

Reviewer 3 Report

 It is important to give more details specifically about the differences in the methods and the obtained products. Energy consumed in each case, possibilities of scaling up for the different methods. And for the zinc nanoparticles, it is necessary to add other characterization techniques, besides XRD and SEM, such as UV absorption and BET (gas absorption) due to different morphology samples can have different surface areas.
Another important issue is to add a brief discussion about how the synthesis method influences the morphology of the ZnO products. 

Author Response

REVIEWER 3 (Round 3)

It is important to give more details specifically about the differences in the methods and the obtained products. Energy consumed in each case, possibilities of scaling up for the different methods. And for the zinc nanoparticles, it is necessary to add other characterization techniques, besides XRD and SEM, such as UV absorption and BET (gas absorption) due to different morphology samples can have different surface areas.

The specific surface area was determined. The results are included in the Table 1.

Another important issue is to add a brief discussion about how the synthesis method influences the morphology of the ZnO products. 

The information was included in the text. FIGURES 7 AND 8

Round 2

Reviewer 1 Report

Dear authors, thank you for your efforts to revise the paper and submit a revised version of the paper bringing it to an average standard allowing its publication. In table 1 I may understand that Ts is the starting temperature and Te the temperature at the end of the process, however, please add a note to the end of this table to describe their meaning.

Author Response

A note describing the meaning of temperature was added below the table.

Reviewer 3 Report

 I have no further comments to make

Author Response

I have no further comments to make

THANK YOU VERY MUCH

Best Regards

Tomasz Strachowski
